# Increasing HPV Vaccination Uptake among Adolescents: A Systematic Review

**DOI:** 10.3390/ijerph17217997

**Published:** 2020-10-30

**Authors:** Anna Acampora, Adriano Grossi, Andrea Barbara, Vittoria Colamesta, Francesco Andrea Causio, Giovanna Elisa Calabrò, Stefania Boccia, Chiara de Waure

**Affiliations:** 1Section of Hygiene, University Department of Life Sciences and Public Health, Università Cattolica del Sacro Cuore, 00168 Rome, Italy; annina.acampora@gmail.com (A.A.); adriano.grossi@yahoo.it (A.G.); andreabarbara89@hotmail.it (A.B.); vittoria.colamesta@gmail.com (V.C.); francescoandreacausio@gmail.com (F.A.C.); stefania.boccia@unicatt.it (S.B.); 2Unità Operativa Complessa Direzione Sanitaria S. Spirito e Nuovo Regina Margherita, Local Health Unit ASL RM1, 00193 Rome, Italy; 3VIHTALI (Value in Health Technology and Academy for Leadership & Innovation), Spin-Off of Università Cattolica del Sacro Cuore, 00168 Rome, Italy; 4Department of Woman and Child Health and Public Health—Public Health Area, Fondazione Policlinico Universitario A. Gemelli IRCCS, 00168 Roma, Italy; 5Department of Experimental Medicine, University of Perugia, 06132 Perugia, Italy; chiara.dewaure@unipg.it

**Keywords:** systematic review, HPV vaccination, increasing coverage

## Abstract

Human Papillomavirus (HPV) vaccination is a well-known fundamental strategy in the prevention of cervical cancer, as it is always caused by HPV infection. In fact, primary prevention of the infection corresponds to primary prevention of HPV-related cancers and other diseases. Since an effective prevention at the population level is the final goal, it is mandatory for healthcare systems to achieve a high HPV vaccination coverage among the adolescents to reduce the circulation of the virus and the burden of HPV-related diseases. This research identified, through a systematic literature review, 38 papers on strategies adopted to increase HPV vaccination coverage among adolescents. The evaluated strategies targeted adolescents/parents and/or healthcare providers and could be grouped in three main types: (1) reminder-based, (2) education, information, and communication activities, and (3) multicomponent strategies. Several types of strategy, such as those relied only on reminders and integrating different interventions, showed a positive impact on vaccination coverage. Nonetheless, the heterogeneity of the interventions suggests the importance to adapt such strategies to the specific national/local contexts to maximize vaccination coverage.

## 1. Introduction

Human papillomavirus (HPV) is one of the most frequent sexually transmitted infections [1]. More than 80% of sexually active women and men are expected to be infected by at least one HPV type by the age of 45 years [2]. In Europe, the prevalence of a detectable HPV infection in women is 14% but varies widely across countries and age groups, with a large peak during adolescence in many countries [1]. In most of the cases, HPV infections are transient and clear up within few months, but sometimes they can persist and progress to cancer. This may happen, in particular, if the infection is due to the so-called high-risk oncogenic HPV genotypes [3]. It is estimated that the vast majority of detected infection (over 70%) is due to these high-risk HPV genotypes, namely, HPV 16, 18, 31, 33, 35, 39, 45, 51, 52, 56, 58, 59, 68, 73, and 82 [4], that are classified as “carcinogenic to humans” by the International Agency for Research on Cancer [5]. These genotypes are known to be responsible for precancerous and cancerous lesions of the anogenital area in both males and females (including cervical, vaginal, vulvar, anal, and penile cancer) and also of other districts such as the head and neck region [6]. In particular, the carcinogenic HPV genotypes 16 and 18 cause about 70% of all cervical cancers (CC) worldwide, and types 31, 33, 45, 52, and 58 cause an additional 20% [7].

CC is the third most common cancer among women worldwide, with 569,847 new cases (13.1% of all cancer in women) and 311,365 deaths (6.9% of all cancer deaths in women) in 2018 [8]. In Europe, the age-standardized incidence rate of CC attributable to HPV in 2018 was 11.2 per 100,000 women [9]. Furthermore, it is estimated that virtually all cases (100%) of CC are attributable to an HPV infection, and this makes it one of the most preventable forms of cancer on a global scale [10].

There are currently three vaccines available to prevent HPV infection and related diseases: a bivalent vaccine, targeting high-risk HPV genotypes 16 and 18, a quadrivalent vaccine including also two low-risk HPV genotypes, namely, HPV 6 and 11 (causing mainly genital warts), and a nonavalent vaccine that counteracts HPV genotypes 6, 11, 16, 18, 31, 33, 45, 5,2 and 58.

According to the World Health Organization (WHO), the primary target of HPV vaccination is represented by girls aged 9–14 years, as achieving high vaccination coverage in girls (>80%) reduces the risk of HPV infection in boys too. Vaccination of secondary target populations, e.g., females aged ≥15 years or males, is recommended only if it is feasible, affordable, cost-effective, and does not divert resources from vaccination of the primary target population or from effective CC screening programs [11].

HPV vaccination has the potential to reduce not only the incidence of CC by as much as 90% [12] but also health care costs, as a consequence of a reduced use of health resources [11].

Despite this, optimal vaccination coverage is hardly achieved, and public health strategies aimed at improving vaccination coverage are required. It is estimated that 47 million women worldwide have been vaccinated against HPV (full course) by 2015, accounting for 1.4 % of the world female population [4]. Among leading countries, Australia has reached HPV vaccination coverage as high as 80.2% in females and 75.9% in males [13].

In Europe, even though most countries belonging to the European Union offer a publicly funded national HPV program, the overall coverage for all doses among the primary target population (including catch-up cohorts) was 52.8% (95% Confidence Interval—95% CI 44.5–61.7) by October 2014 [4]. Vaccination coverage varies considerably across Europe, not only between countries but also within the same country. Finland, Iceland, Norway, Spain, Sweden, and the UK, for example, have reported national coverage higher than 70%, while in other countries, such as France and Germany, coverage has been significantly lower than 50% [14].

In Italy, the current National Immunization Plan 2017–2019 actively offers free vaccination to girls and boys aged 11 years and sets a 95% target vaccination coverage by 2020 [15]. As far as the schedule is concerned, the number of doses depends on patient’s age and on the vaccine but, generally speaking, vaccination is offered in two doses to girls and boys in their 12th year of age [16]. Despite the efforts to reach the goal of 95%, the last Italian available data showed that vaccination coverage was way below the target: first-dose and all-doses coverage were 61.7% and 40.3% in the 2006 female birth cohort and 24.1% and 19.3% in the 2006 male birth cohort. Nevertheless, there was a significant variability among different regions and autonomous provinces and among birth cohorts, with all-doses coverage ranging from 70.1% among girls born in 1999 to 62.2% among girls born in 2005 [17].

Several types of strategies to increase vaccination coverage in general or for specific vaccines such as the one against HPV have been already described in the scientific literature [18,19,20,21,22,23]. Some authors [18,19,22] reviewed strategies to improve adolescents’ vaccination coverage for different types of vaccine or the effect of specific interventions, namely, patient reminders and recall systems, to improve immunization rates in the whole population [23]. Other papers specifically focused on HPV, investigating strategies targeted to both adolescents and younger adults (11–26 years) [20,21,24].

The present research is aimed at contributing to the existing body of evidence focusing only on adolescents as the primary target of HPV vaccination in high-income countries. The objective of this systematic review is therefore to gather evidence from the existing literature concerning strategies that can be used in high-income countries to increase HPV vaccination coverage in the adolescent population. In particular, the objectives of this systematic review are to identify the main types of strategies that can be adopted and to assess their impact on vaccination coverage.

## 2. Materials and Methods

In order to collect scientific evidence related to the aim of the study, a systematic review of the literature was performed through the consultation of scientific databases and grey literature. The results are reported according to PRISMA (Preferred Reporting Items for Systematic Review and Metanalysis) guidance [25].

The Population Intervention Comparison Outcome (PICO) model used for the search of evidence was as follows.

(P) The population addressed in this systematic review included the different possible targets of strategies aimed at increasing vaccination coverage in adolescents, including adolescents themselves, their parents, and healthcare professionals involved in vaccination services.(I) Every kind of strategy implemented in order to increase HPV vaccination coverage among the adolescent population (9–18 years) according to the WHO definition of adolescents [26] and the WHO recommendation of HPV-vaccination was included [27].(C) The usual strategy, namely, the absence of any specific strategy aimed at increasing coverage was included.(O) HPV vaccination coverage in terms of vaccination initiation and/or all doses completion was considered.

### 2.1. Search Strategy

PubMed, Scopus, and Web of Science were used for searching articles. The following keywords and synonyms, linked by Boolean operators, were used: papillomavirus vaccines, HPV, papillomaviridae, vaccination, immunization program, vaccination practice, immunization practice, vaccination strategy/ies, immunization strategy/ies, intervention, immunization, effectiveness, efficacy, efficiency feasibility, vaccination coverage, implementation, evaluation, assessment, impact. An additional search was carried out through the screening of the references of the included studies.

The last update of the search was conducted on 17 July 2018 without temporal limits, and the records found were entered in a Microsoft Excel sheet. Firstly, a check for duplicates was performed, and unique records were subsequently screened for inclusion/exclusion.

### 2.2. Inclusion/Exclusion Criteria

All the articles involving humans and written in English or Italian were included in the review if matching the PICO and conducted in high-income countries [28].

Studies focused on immunocompromised patients only (e.g., HIV-positive individuals) were excluded, as well as studies on patients and/or healthcare providers’ knowledge, attitudes, and behaviors, possible factors/determinants of vaccination uptake, and legislative interventions. Other reasons for exclusion were the impossibility to retrieve the full text and the type of article (systematic reviews or opinion/position papers, guidelines, editorials, or commentary were excluded).

### 2.3. Selection Process and Data Extraction

The screening process was conducted by reading first the titles and abstracts and then the full texts. Three Authors (A.A., A.G., A.B.) independently carried out the selection process and extracted the data. Any disagreement was resolved by discussion or by asking a fourth author (C.d.W.) until a consensus was reached. After the definition of the final list of eligible full-text articles, data were collected and entered in a spreadsheet. The following information was extracted: general information (first author, year of publication, intervention (strategy), country of implementation of the strategy, target of the strategy (population), study design, study size, study outcomes, main results).

The heterogeneity of the results did not allow to perform a meta-analysis; therefore, a narrative synthesis was used to synthetize the results. Strategies were considered effective when a significant increase in HPV vaccination coverage was reported.

## 3. Results

The bibliographic search yielded 2164 citations on PubMed, 2303 on Scopus, and 1923 on Web of Science. Of them, 2819 were removed because duplicates, leaving 3571 single citations. After title and abstract screening, 3420 articles were discarded. According to the pre-specified inclusion and exclusion criteria, 33 articles were included in the qualitative synthesis. Five additional studies were found by examining the references included in the articles. At the end of the selection process, 38 articles [27,28,29,30,31,32,33,34,35,36,37,38,39,40,41,42,43,44,45,46,47,48,49,50,51,52,53,54,55,56,57,58,59,60,61,62,63,64] were included in the qualitative synthesis (Figure 1).

### 3.1. Characteristics of the Studies

Appendix A shows the main characteristics of the included studies, which were published between 2011 and 2018. Most of them (31; 81.6%) were conducted in the USA, six (15.8%) in Europe (Sweden, England, Italy, Netherlands, and Scotland), and one (2.6%) in Japan. The study population ranged from 19 [29] to 325,229 [30] individuals.

Many of the studies were randomized control trials (RCT) (17; 44.7%) or had a quasi-experimental (QE) design (13; 34.2%). The remaining were cohort studies (4; 10.5%), cross-sectional studies (2; 5.3%), one non-randomized trial (NRT), and one ecologic study (ES) (1 for both; 2.6%). All but four studies (89.5%) were based on registries, with the remaining ones based on self-reported data.

### 3.2. Description of Identified Strategies and Synthesis of the Results

The evaluated strategies were very heterogeneous according to both the type and the number of interventions adopted. In order to synthetize the results, the identified strategies were grouped in three main types:
(1)reminder-based interventions,(2)education, information, and communication strategies,(3)multicomponent interventions.

The last category included all the strategies based on two or more types of interventions (e.g., reminder, information, education, financial incentives, feedback for providers, school-based intervention). Furthermore, also the targets of the strategies were grouped as follows: adolescents and/or their parents (A/P), health care providers (HCPs), or both (A/P; HCPs). Regarding the outcomes, seven studies (18.4%) evaluated vaccination initiation, three studies (7.9%) analyzed only all-doses completion, while 27 studies (71%) reported both initiation and all-doses completion. Eventually, one study reported data on each dose (2.6%) coverage.

In order to report a synthesis of the results, data were grouped together considering the type of strategy, its target, and the reported outcome(s). A synthesis of the proven efficacy of the strategies in shown in Table 1. A strategy was considered effective if a significant increase was reported at least in one of the study’s outcomes, i.e., vaccination initiation and/or all-doses completion (Table 1).

#### 3.2.1. Reminder-Based Strategies

Six studies (15.8%) analyzed reminder-based strategies. Five of these targeted A/P, in all cases with an RCT design, while one cohort study targeted HCPs. The reminders sent to A/P generally consisted of text messages, telephone calls, letters, or e-mails. Three studies evaluated vaccination initiation as an outcome, and in all cases (3/3; 100%), a significant increase was reported [32,34,35], while an increase in all-doses completion was found in two of three (67%) studies [33,34] that analyzed this outcome. No improvements were noted by Kempe et al., in all-doses completion [31].

The reminders for HCPs were automated electronic alerts that were activated at any visit in case of missed appointments or in the presence of a subject who was not yet vaccinated and showed significant positive results in terms of both vaccination initiation and doses completion [36].

#### 3.2.2. Education, Information and Communication Strategies

Eleven studies (28.9%) described the implementation of education, information, and/or communication strategies. Four of these (36.4%) targeted both A/P and HCPs, and seven (63.6%) targeted A/P alone. The latter were very variegated and foresaw the delivery of short messages with rhetorical questions (a technic called “foot in the door”) [40], the distribution of educational brochures with an invitation to group health education [38], gender-specific postcards inviting parents to discuss HPV vaccination with their HCPs [41], face-to-face structured information in schools [37], or social events with power point presentation [42]. A more complex intervention was developed by Lee et al., and consisted in storytelling narrative videos focused on HPV vaccination [29], while Pot et al., developed a website providing personalized information based on preferences and needs [39]. The results obtained by this kind of strategy were very heterogeneous. All studies analyzed the first dose uptake, but just one reported significant results when gender-specific postcards inviting parents to discuss HPV vaccination with their HCPs were used [41]. In the same way, also the efficacy for all-doses completion was demonstrated in one study [38] in which a brochure was provided together with group health education, referral, and navigation support from a trained community health worker and from student peer educators.

Three out of four studies targeting both A/P and HCPs were published by the same authors. Two of these described the implementation of a social marketing campaign based on four principles: product (recommend vaccine against HPV), price (cost, perception of safety and efficacy, and access), promotion (posters, brochures, website, news releases, doctor’s recommendation), and place (doctor’s offices, retail outlets) [43,44,45]. These studies reported discordant results. The third study, instead, demonstrated the efficacy of focus groups with providers, parents, and preteens, in addition to an online training, in increasing both initiation and all-doses completion among 140,000 individuals, [45]. An additional study [46] evaluated an intervention including the delivery of personalized educational videos for adolescents with the possibility to ask questions to team researchers and a one-hour training session for providers. The vaccination initiation did not increase significantly, and a lower probability of all-doses completion was reported after the intervention.

#### 3.2.3. Multicomponent Interventions

The major part of the included studies (*n* = 21; 55.3%) evaluated multicomponent strategies. Out of these, nine studies (42.8%) targeted A/P alone, six (28.6%) HCPs alone, and six (28.6%) both A/P and HCPs.

Multicomponent strategies were widely heterogeneous, and this makes it very difficult to perform a synthesis. Nevertheless, multicomponent strategies often included information/education interventions (16; 76.2%) [30,47,50,51,52,54,56,57,58,59,60,61,62,63,65,66] and reminders (14; 66.7%) [47,48,49,50,51,52,55,59,60,61,62,63,64,66]; the latter were combined in 10 cases (47.6%) [47,50,51,52,59,60,61,62,63,66]. Other interventions included feedback reports on coverage (8; 38.1%) [55,56,57,58,60,61,62,63], incentives (4; 19%) [48,49,65], and school-based strategies (4; 19%) [30,48,51,53].

Four out of nine (44.4%) multicomponent strategies targeting A/P included an information/education intervention associated with one or more reminder activities [47,50,51,52]. Two studies (22.2%) included a reminder intervention and a financial incentive for adherence [48,49], and four studies (44.4%) relied also on school-based interventions [30,48,51,53] including the provision of vaccination at school in one case [30]. The study by Zimmerman et al., assessed a strategy called “Four Pillars Transformation Program Practice” that combined communication activities with parents and adolescents with interventions aimed at facilitating access to vaccination [54]. In general, all these studies described an increase in vaccination coverage in terms of initiation and/or all-doses completion. Nevertheless, a significant effect was found in five out of eight studies (62.5%) evaluating HPV vaccination initiation [47,48,49,52,54] and in three out of eight works (37.5%) reporting all-doses completion [47,48,49]. The ninth study [30] demonstrated the efficacy of a school-based information program together with school-based vaccination for any administered vaccine dose.

Multicomponent strategies targeting HCPs mostly included educational and training interventions (5; 83.3%) [56,57,58,59,60] and periodic feedback reports on coverage (5; 83.3%) [55,56,57,58,60], followed by reminders (3; 50%) [55,59,60]. Two studies (33.3%) reported no efficacy [55,57], whereas the remaining four (66.7%) showed a significant impact in terms of either HPV vaccination initiation and all-doses completion [56,58,59,60].

Multicomponent strategies targeting both A/P and HCPs mostly included educational and training interventions (5; 83.3%) [61,62,63,65,66] and reminders (5; 83.3%) [61,62,63,64,66], followed by periodic feedback reports on coverage (3; 50%) [61,62,63]. All described positive effects in terms of both HPV initiation and completion [61,62,63,64,65,66], but only three of them (50%) reported significant results [63,64,65]. In particular, offering HPV vaccination in bundle with other vaccinations (tetanus toxoid, reduced diptheria toxoid, acellular pertussis—Tdap- and tetravalen meningococcal—MCV4) to all subjects attending a medical visit, together with the establishment of an internal immunization registry and standing orders/reminders before any visit, showed to increase both first-dose uptake and all-doses completion [64]. Significant increases in coverage were also obtained by combining provider education, periodic feedback reports on coverage, and reminders intended for patients [63] as well as with the provision of information and education interventions for providers and A/P associated with economic incentives [65].

## 4. Discussion

This systematic review was aimed at gathering available evidence on strategies to increase HPV vaccination coverage among adolescents. According to the reviewed literature, it can be agreed that different types of strategies might have a positive influence on HPV vaccination coverage, in terms of both vaccination initiation and completion.

Although the largest amount of literature on the topic is from the USA, other countries with different healthcare systems have also performed studies. This conveys the importance of HPV vaccination from the viewpoint of public health worldwide.

Strategies identified often consisted of one or more interventions including reminders, education and training, information delivery and communication campaigns, feedbacks on coverage data, and other interventions such as financial incentives and school-based interventions.

Reminder-based strategies, either alone or in combination with other interventions, got positive results in several studies in particular in regard to vaccination initiation. Furthermore, these strategies showed positive results when addressed to both vaccination targets or their parents and HCPs. Furthermore, the evidence on reminder-based strategies came mostly from RCT, and this makes the overall results quite reliable. Interestingly, as reported by Kempe et al. [31], although no difference in all-doses completion was reported between intervention and control groups, coverage was higher for those receiving e-mail or telephonic reminders (90% vs. 60%, *p* = 0.008) when compared to those informed by other means, like text messages. In addition, some studies reported that while a telephone call seems to be effective in increasing vaccination initiation, text messages were more effective in increasing all-doses completion [32,33]. Different preferences were also reported among adolescents and their parents. While adolescents seem to prefer text messages and e-mail, their parents seem to prefer postcard reminders, whereas telephone calls are described as the least effective tool [18]. These observations suggest a potential different impact based on the tools adopted to send reminders, but further research should investigate this aspect.

The evidence about reminder-based strategies targeting HCPs came from a single big cohort study [36]. Anyway, in most of the cases, reminders targeting HCPs were included in multicomponent interventions. However, it should be noticed that this kind of strategy often requires electronical devices that might not have been implemented in all healthcare structures. This might limit the feasibility of their use in some setting.

In general, reminders were shown to improve the immunization rate among adolescents [23], but previous evidence also reported a differential effect depending on the outcomes, as reminders for A/P seemed to be more effective on vaccination initiation while those for HCPs were associated with improvement in all-doses completion [20]. This observation was confirmed in our review, since reminders intended for A/P showed more often a positive impact on vaccination initiation.

Communication, information, and education strategies are certainly important in public health, especially in the socio-cultural context of Western countries, where misinformation, ethical/religious issues, and conspiracy theories are undermining people’s trust in vaccines. Nonetheless, these strategies were shown to be the least effective in increasing HPV vaccination coverage when implemented alone and if directed only to A/P.

As described also by other authors [67], promising results were found for strategies implementing peer-to-peer education [38], focus group involving both patients and providers [45], and group education [38,45]. A positive impact was also suggested by a pilot testing of an articulated storytelling narrative video on HPV vaccination based on the revised Network Episode Model in which, although there was no difference in vaccine initiation between intervention and control groups, preliminary results showed a higher intention to receive HPV vaccination in the intervention group [29]. Nonetheless, the evidence on education campaigns, implemented either at individual or at community level, generally showed that the improvement in vaccination coverage was limited to the intervention period and did not persist after the end of the intervention [20].

On the contrary, combining education, information, or communication activities with other kinds of interventions seems to be more effective and durable. In fact, in general, multicomponent strategies showed the best results: this might be explained by the fact that being multi-faceted, these strategies can be better adapted to the environment and the target.

Most studies reporting positive results of multicomponent strategies intended for A/P included at least a reminder [47,48,49,50,52], often together with an informative intervention [47,50,52]. Similarly, effective strategies focused on HCPs evaluated at least provider education (either about updated evidence on HPV infection/vaccination or about communication strategies with parents on HPV vaccination) and training [56,59,60] associated with a form of reminder, such as an electronic alert on non-attenders or patients late for vaccination [59,60] and/or individualized feedback on vaccination coverage [56,58,60].

Focusing on data about feedback on vaccination coverage, it has been previously recognized that giving providers an assessment and feedback report on their performance in achieving vaccination coverage improves their motivation and, consequently, coverage. This approach, indeed, was also recommended by the Community Preventive Services Task Force (CPSTF) of the Center for Disease Control and Prevention (CDC) that indicated the AFIX (Assessment, Feedback, Incentives, and Exchange) program [68] as an effective strategy to improve vaccination coverage in general. Nowadays, the program has been replaced by the Immunization Quality Improvement for Providers (IQIP) program, which is based on scheduling the following immunization visit before the patient leaves the provider site, leveraging immunization information system functionality to improve immunization practice, and giving strong vaccine recommendations (including on HPV vaccination if the provider has adolescent patients) [69].

Studies evaluating strategies with multicomponent interventions targeting both A/P and HPCs also showed positive results in many cases. At the same time, Fujiwara et al., demonstrated that the higher the number of interventions implemented within a strategy, the greater the achieved HPV vaccination coverage, with the largest one found with a school-based vaccination program associated with a public subsidy [48].

It should be mentioned that some multicomponent interventions implemented at school level were proven to be advantageous [48,51,53]. In Fujiwara et al. [48], in particular, a school-based vaccination program was reported to be the most effective strategy when compared with different combinations of other interventions including economic incentives, reminders, or no strategies. The study by Vanderpool et al., described a large effect of the strategy that included a school-based intervention with provision of educational materials to parents, telephonic contacts to inform them about the program and to remind vaccination appointments, information materials for students (articles, website, t-shirt, school events as pizza parties, etc.), and active students’ involvement in materials development [51]. Another relevant aspect was that school-based interventions including vaccination at school and school personnel engagement increased vaccination coverage [53], while school-based information/education interventions alone showed discordant results [30,40].

Besides the type of implemented strategies, further factors could influence adolescents’ vaccination adherence, including perception of the personal risk of infection and disease, vaccine acceptability, perceived vaccine benefits and risks, and, not least, parents’ attitudes and cultural background. Furthermore, depending on the specific context, also financial barriers could play an important role, for example in the USA [70,71].

Indeed, the existence of several context-specific factors requires that strategies to increase HPV vaccination coverage should be developed and adapted based on the national/local context. Anyway, our review suggests that the chosen strategy should be multicomponent, and this result is also confirmed by previous papers.

In fact, even though implementing a single strategy showed some positive results, previous reviews performed in the USA, aimed at identifying strategies to increase coverage in general [18,21] and for HPV vaccine in particular [18,21], suggested that interventions could be more successful when combined.

In particular, environmental interventions, such as school-based programs, were identified as fundamental for reaching a large target population and overcoming potential barriers to the access to healthcare [20]. In Australia, where high HPV vaccination coverage has been achieved, vaccination is provided through school-based immunization programs.

Also, according to a recent Cochrane review [22] published at the beginning of 2020, multicomponent interventions can improve the uptake of HPV vaccine compared to usual practice. This review [22] focused on interventions to improve immunization rates for all types of vaccination in adolescents, including HPV. In comparison to our review, Abdullahi et al., applied different and more strict eligibility criteria especially regarding study design, including only randomized trials, non-randomized trials, interrupted time-series studies, and controlled before–after studies meeting quality criteria used by the Cochrane Effective Practice and Organisation of Care. Eventually, a lower number of studies (11 studies) focused on HPV vaccination was included as compared to our review (38 studies). Furthermore, the Cochrane review focused on strategies other than reminders and took into account countries with all levels of income. Also, the approach to the classification of interventions was slightly different from ours, as the authors first distinguished among recipient-oriented, provider-oriented, and health system interventions. Indeed, we believe that the two reviews together could give a complementary picture of effective interventions to improve HPV vaccination coverage.

Although this is not a specific focus of our study, an additional point needs to be considered with respect to HPV vaccination of boys, because of the lower vaccination coverage observed for them as compared to girls. In fact, a recent study highlighted the importance to have a gender-neutral approach to pursue HPV oncogenic genotypes eradication, which, in fact, might be achieved even with moderate vaccination coverage [72]. Indeed, future research should explore the possible differences of strategies for girls and boys. In this respect, also our review showed some differences in term of the impact on HPV vaccination coverage between males and females. Nevertheless, our results were not conclusive, as some papers showed better results in males [44,58,60,64] and vice versa [41,59,66].

The main limitation of the present systematic review is that no formal quality assessment of the included studies was performed because of study design heterogeneity and the fact that a quantitative analysis was not carried out. Nevertheless, in order to perform a critical analysis and a synthesis of the results, the study design and size were considered. Another limitation could be the difference in healthcare systems across countries but, on the other hand, this could also be a strength, since it allowed providing a more comprehensive overview of strategies settled in high-income countries independently from the healthcare organization.

The strengths of the study include the robustness of the screening and data extraction processes performed by three authors and the use of three databases for the literature search, which made this systematic review exhaustive.

## 5. Conclusions

Improving HPV vaccination coverage in adolescents is a relevant public health goal that must be pursued in order to reduce the incidence of HPV-related diseases. This review showed that, except for reminder-based strategies targeting adolescents and/or their parents, single strategies are often sufficient to increase vaccination coverage, being multicomponent interventions the best ones. Public health systems should therefore work on the integration of several approaches, including personalized reminders, information and education activities aimed at increasing adolescents’, parents’, and HCPs’ awareness and knowledge about HPV infection and vaccine, HCPs training on communication strategies with parents and adolescents, and facilitation of the access to vaccination also through school-based vaccination programs. Further research could be useful to clarify individual and contextual factors that could modify the efficacy of these strategies.

## Figures and Tables

**Figure 1 ijerph-17-07997-f001:**
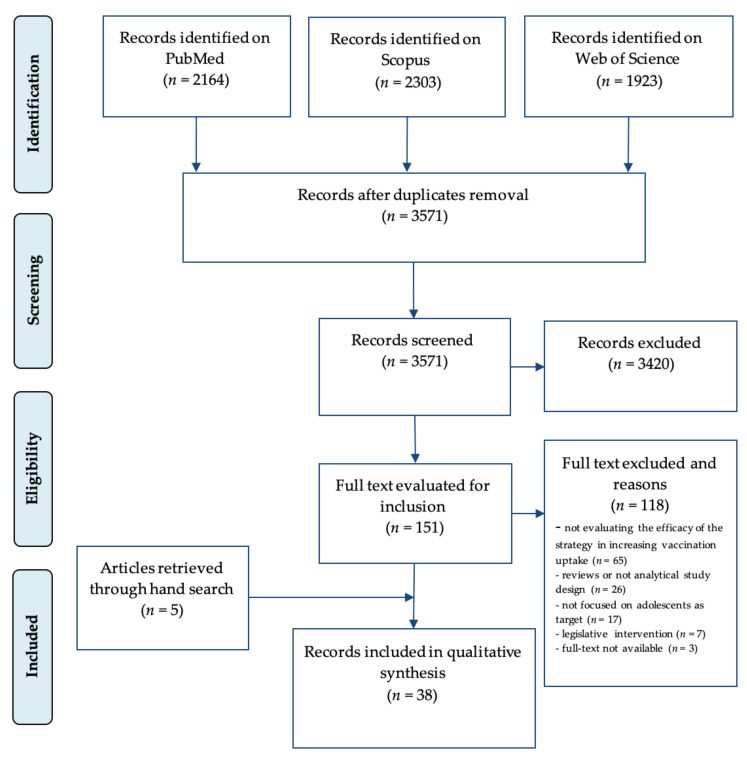
Flow chart of article selection process. Modified from: Moher D, Liberati A, Tetzlaff J, Altman DG, The PRISMA Group (2009). Preferred Reporting Items for Systematic Reviews and Meta-Analyses: The PRISMA Statement. BMJ 2009; 339:b2535, doi: 10.1136/bmj.b2535 [23].

**Table 1 ijerph-17-07997-t001:** Results synthesis.

Target	Number of Studies	Study Design	Size (Range)	Proven Efficacy of the Strategy *
Reminder-based strategies (*n* = 6)				
A/P	5	5 RCT	374–7546	4/5
	(31–35)			(80%)
HCP	1	1 CS	15,021	1/1
	(36)			(100%)
Education/Information and Communication strategies (*n* = 11)				
A/P	7	5 RCT; 2 Q-E	19–8062	2/7
	(29,37–42)			(28.6%)
A/P; HCP	4	2 Q-E; 1 C-S;	225–25,869	2/4
	(43–46)	1 NRT		(50%)
Multicomponent interventions (*n* = 21)				
A/P	9	3 RCT; 2 Q-E;	53–325,229	7/9
		2 CS; 1 C-S		
	(30,47–54)			(77.8%)
		1 Ecologic		
HCP	6	3 RCT; 3 Q-E	50–107,443	4/6
	(55–60)		(1 nr)	(66.7%)
A/P; HCP	6	4 Q-E; 1 RCT;	105–16,041	3/6
	(61–66)	1 CS	(1 nr)	(50%)

Abbreviation: A/P Adolescents/Parents; HCP HealthCare Providers; RCT Randomized Controlled Trial; NRT Non-Randomized Trial; CS Cohort Study; C-S Cross-Sectional Study; Q-E Quasi-Experimental; nr Not Reported; * Significant increase in vaccination initiation and/or all-doses completion.

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
