# Peer review of "Increasing HPV Vaccination Uptake among Adolescents: A Systematic Review"

_ijerph, 2020, doi:10.3390/ijerph17217997_

Round 1
Reviewer 1 Report
Apart from a typo in line 362, generally a good paper.
Author Response
Thanks to the Reviewer for the comment. We replaced “whit the exception” with “except for”, now in line 440
Reviewer 2 Report
This systematic review was aimed at gathering available evidence on strategies to increase HPV 258 vaccination coverage among adolescents, in terms of both vaccination initiation and completion. This is an interesting paper.
In the introduction, when you talk about Italy add the vaccination plan. Consult and add the following paper to the references: PMID: 32630772.
In the Materials you should report methods used for assessing risk of bias of individual studies (including specification of whether this was done at the study or outcome level), and how this information is to be used in any data synthesis. English can be improved.
Author Response
Point 1. In the introduction, when you talk about Italy add the vaccination plan. Consult and add the following paper to the references: PMID: 32630772.
Response 1. We thank the Reviewer for these suggestions. We specified the vaccination schedule at line 90-91 and added the suggested paper to references (16).
Point 2. In the Materials you should report methods used for assessing risk of bias of individual studies (including specification of whether this was done at the study or outcome level), and how this information is to be used in any data synthesis.
Response 2. Thanks to the Reviewer to have highlighted this point. As we have better clarified in the final part of the discussion, the main limitation of this work is that we did not perform a formal quality appraisal of included studies because of study design heterogeneity and the fact that a quantitative analysis was not carried out (lines 426-430). In fact, without any quantitative synthesis it becomes quite difficult to address the impact of the quality of studies on overall results. Nevertheless, we considered study design and size in making the overview of results.
Point 3. English can be improved.
Response 3. Thanks for the suggestion. We revised the English throughout the text.
Reviewer 3 Report
Acampora et al. have presented a manuscript entitled ´ Increasing HPV vaccination uptake among adolescents: a systematic review ´ to be considered for publication in the International journal of Environmental Research and Public Health.
Presented review article has a timely topic regarding HPV vaccine strategies to enhance vaccine coverage, however, several caveats needs to be covered before the review is comprehensive and can be published.
The main issue is that the review completely ignores the fundamental aspect of boys HPV vaccination, which is previously shown by Vänskä et al. 2020 JID, to be of utmost important, if WHO aims to reach its goal of cervical cancer eradication. Vaccine high coverage is not reachable at global level, hence, gender-neutral strategy will play an important role, if enough vaccines are available.
In summary, the manuscript needs an updated literature review and a major revision for being more concise and clear with the results and specifically for discussion to be within the context of previous reports regarding gender-neutral strategies.
Reference:
Vänskä et al. Vaccination With Moderate Coverage Eradicates Oncogenic Human Papillomaviruses If a Gender-Neutral Strategy Is Applied. J Infect Dis. 2020 17;222(6):948-956.
Author Response
Point 1. The main issue is that the review completely ignores the fundamental aspect of boys HPV vaccination, which is previously shown by Vänskä et al. 2020 JID, to be of utmost important, if WHO aims to reach its goal of cervical cancer eradication. Vaccine high coverage is not reachable at global level, hence, gender-neutral strategy will play an important role, if enough vaccines are available.
In summary, the manuscript needs an updated literature review and a major revision for being more concise and clear with the results and specifically for discussion to be within the context of previous reports regarding gender-neutral strategies.
Response 1. We would like to thank the Reviewer for rising this important point and for the suggested reference. As already specified in the introduction at lines 64-66, “according to the World Health Organization (WHO), the primary target of HPV vaccination is girls aged 9–14 years as achieving high vaccination coverage in girls (>80%) reduces the risk of HPV infection for boys too”. Nevertheless, we did not focus our review only on girls but on the whole adolescent population and, in fact, few papers included also released gender specific results. As we recognize the importance of the highlighted point, we tried to better address the issue in discussion synthetizing also the evidence coming from studies included in the review that provided gender specific results (lines 419-427).
Reference:
Vänskä et al. Vaccination With Moderate Coverage Eradicates Oncogenic Human Papillomaviruses If a Gender-Neutral Strategy Is Applied. J Infect Dis. 2020 17;222(6):948-956.
Reviewer 4 Report
I appreciate the opportunity to review this manuscript by Anna Acampora et al. This review article studied the impact of different strategies on HPV vaccination and concluded that different types of strategies all have a positive influence on the HPV vaccination coverage. Overall, this manuscript is well crafted. The authors presented a nice explanation of human papillomavirus, HPV vaccine, and worldwide HPV vaccination coverage in the section of the introduction. Literature mining and the article selection procedure were well described. A tidy summary of included publications was provided in the supplementary as well. I have a few trivial comments which are listed below:
(1) The strategy category of “multicomponent interventions” remains somewhat confusing to readers. Would it be possible to provide an additional statistical table with detailed strategy break downs for this particular category? That would be very helpful for the data interpretation.
(2) Though it is not the focus of the review, I would suggest the authors discuss if different types of HPV vaccines were used among all included studies. If yes, I am wondering if there is an association between the HPV vaccine type and the coverage?
(3) From lines 91-95, the paragraph mentioned that, several strategies have been reported to increase HPV vaccination coverage. Moreover, the authors also acknowledged that the impact of HPV strategies on both adolescents and younger adults has been presented elsewhere. This particular is too abstract. The authors should carefully discuss, what are the conclusions from these similar review studies? Does this review article identify consistent/distinct conclusions when compared to citation 17 to 21?
(4) Though without any available reference, it would be interesting to briefly discuss, what strategies have been employed by Australia’s leadership to achieve the 80.2% vaccination coverage?
Author Response
Point 1. The strategy category of “multicomponent interventions” remains somewhat confusing to readers. Would it be possible to provide an additional statistical table with detailed strategy break downs for this particular category? That would be very helpful for the data interpretation.
Response 1. Thanks to the Reviewer for the suggestion. We revised the whole 3.2.3. section in order to provide additional information on type of interventions included in the multicomponent strategies. We did not include a specific table as table 1 already describe the component of each strategy. In the review process we also corrected a refuse in table 1.
Point 2. Though it is not the focus of the review, I would suggest the authors discuss if different types of HPV vaccines were used among all included studies. If yes, I am wondering if there is an association between the HPV vaccine type and the coverage?
Response 2. We thank the Reviewer for this suggestion. We believe that this kind of information could be somehow confusing because we should also take into consideration the time of the introduction of vaccination and specific and general trends in vaccination coverage. Furthermore, our aim was to identify strategies useful to increase vaccination coverage independently by the type of vaccine also because the review was carried out at worldwide level. Indeed, we would like to not address this point.
Point 3. From lines 91-95, the paragraph mentioned that, several strategies have been reported to increase HPV vaccination coverage. Moreover, the authors also acknowledged that the impact of HPV strategies on both adolescents and younger adults has been presented elsewhere. This particular is too abstract. The authors should carefully discuss, what are the conclusions from these similar review studies? Does this review article identify consistent/distinct conclusions when compared to citation 17 to 21?
Response 3. Thanks to the Reviewer for this comment. We added to the discussion section more details about results from previous reviews in comparison with our (line 407-416) and we removed from the introduction the generic reference to them.
Point 4. Though without any available reference, it would be interesting to briefly discuss, what strategies have been employed by Australia’s leadership to achieve the 80.2% vaccination coverage?
Response 4. Thanks to the Reviewer for the interesting suggestion. We referred to school-based immunisation programs for HPV vaccination in Australia in the discussion (lines 415-416).
Reviewer 5 Report
This review evaluated the Coverage of HPV vaccine among adolescents in a set of developed high-income countries. They examine the different strategies that were followed for better HPV vaccination coverage among adolescents, such as strategies involving reminders, education and training, information delivery and communication campaigns, feed backs for providers, and other interventions such as financial incentives and school-based interventions. They found that reminder-based strategies yield better results. Education and information regarding HPV vaccination were another big winner.
Minor Comment:
1) Need some editorial and format improvements, including English and reference organization in the text.
Author Response
Point 1. Need some editorial and format improvements, including English and reference organization in the text.
Response 1. We would like to thank the Reviewer for the comments. We revised the manuscript and made editorial and format revisions. We also revised English throughout the text.